# Spherical Text Embedding

**Yu Meng**[1], **Jiaxin Huang**[1], **Guangyuan Wang**[1], **Chao Zhang**[2],
**Honglei Zhuang**[1]*,**Lance Kaplan**[3], **Jiawei Han**[1]
[1] Department of Computer Science, University of Illinois at Urbana-Champaign
[2] College of Computing, Georgia Institute of Technology
[3] U.S. Army Research Laboratory
[1] {yumeng5,jiaxinh3,gwang10,hzhuang3,hanj}@illinois.edu
[2] chaozhang@gatech.edu   [3] lance.m.kaplan.civ@mail.mil

## Abstract

Unsupervised text embedding has shown great power in a wide range of NLP tasks. While text embeddings are typically learned in the Euclidean space, directional similarity is often more effective in tasks such as word similarity and document clustering, which creates a gap between the training stage and usage stage of text embedding. To close this gap, we propose a spherical generative model based on which unsupervised word and paragraph embeddings are jointly learned. To learn text embeddings in the spherical space, we develop an efficient optimization algorithm with convergence guarantee based on Riemannian optimization. Our model enjoys high efficiency and achieves state-of-the-art performances on various text embedding tasks including word similarity and document clustering.

## 1  Introduction

Recent years have witnessed enormous success of unsupervised text embedding techniques [29, 30, 33] in various natural language processing and text mining tasks. Such techniques capture the semantics of textual units (*e.g.*, words, paragraphs) via learning low-dimensional distributed representations in an unsupervised way, which can be either directly used as feature representations or further fine-tuned with training data from downstream supervised tasks. Notably, the popular Word2Vec method [29, 30] learns word embeddings in the Euclidean space, by modeling local word co-occurrences in the corpus. This strategy has later been extended to obtain embeddings of other textual units such as sentences [2, 19] and paragraphs [22].

Despite the success of unsupervised text embedding techniques, an intriguing gap exists between the training procedure and the practical usages of the learned embeddings. While the embeddings are learned in the Euclidean space, it is often the directional similarity between word vectors that captures word semantics more effectively. Across a wide range of word similarity and document clustering tasks [3, 16, 23], it is common practice to either use cosine similarity as the similarity metric or first normalize word and document vectors before computing textual similarities. Current procedures of training text embeddings in the Euclidean space and using their similarities in the spherical space is clearly suboptimal. After projecting the embedding from Euclidean space to spherical space, the optimal solution to the loss function in the original space may not remain optimal in the new space.

In this work, we propose a method that learns spherical text embeddings in an unsupervised way. In contrast to existing techniques that learn text embeddings in the Euclidean space and use normalization as a post-processing step, we directly learn text embeddings in a spherical space by imposing unit-norm constraints on embeddings. Specifically, we define a two-step generative model on the surface of a unit sphere: A word is first generated according to the semantics of the paragraph, and then

the surrounding words are generated in consistency with the center word's semantics. We cast the learning of the generative model as an optimization problem and propose an efficient Riemannian optimization procedure to learn spherical text embeddings.

Another major advantage of our spherical text embedding model is that it can jointly learn word embeddings and paragraph embeddings. This property naturally stems from our two-step generative process, where the generation of a word is dependent on its belonging paragraph with a von Mishes-Fisher distribution in the spherical space. Explicitly modeling the generative relationships between words and their belonging paragraphs allows paragraph embeddings to be directly obtained during the training stage. Furthermore, it allows the model to learn better word embeddings by jointly exploiting word-word and word-paragraph co-occurrence statistics; this is distinct from existing word embedding techniques that learn word embeddings only based on word co-occurrences [5, 29, 30, 33] in the corpus.

**Contributions.** (1) We propose to learn text embeddings in the spherical space which addresses the mismatch issue between training and using embeddings of previous Euclidean embedding models; (2) We propose a two-step generative model that jointly learns unsupervised word and paragraph embeddings by exploiting word-word and word-paragraph co-occurrence statistics; (3) We develop an efficient optimization algorithm in the spherical space with convergence guarantee; (4) Our model achieves state-of-the-art performances on various text embedding applications.

## 2 Related Work

### 2.1 Text Embedding

Most unsupervised text embedding models such as [5, 19, 22, 29, 30, 33, 37, 42] are trained in the Euclidean space. The embeddings are trained to capture semantic similarity of textual units based on co-occurrence statistics, and demonstrate effectiveness on various text semantics tasks such as named entity recognition [21], text classification [18, 27, 28, 38] and machine translation [8]. Recently, non-Euclidean embedding space has been explored for learning specific structural representations. Poincaré [11, 31, 39], Lorentz [32] and hyperbolic cone [15] models have proven successful on learning hierarchical representations in a hyperbolic space for tasks such as lexical entailment and link prediction. Our model also learns unsupervised text embeddings in a non-Euclidean space, but still for general text embedding applications including word similarity and document clustering.

### 2.2 Spherical Space Models

Previous works have shown that the spherical space is a superior choice for tasks focusing on directional similarity. For example, normalizing document tf-idf vectors is common practice when used as features for document clustering and classification, which helps regularize the vector against the document length and leads to better document clustering performance [3, 16]. Spherical generative modeling [4, 43, 44] models the distribution of words on the unit sphere, motivated by the effectiveness of directional metrics over word embeddings. Recently, spherical models also show great effectiveness in deep learning. Spherical normalization [24] on the input leads to easier optimization, faster convergence and better accuracy of neural networks. Also, a spherical loss function can be used to replace the conventional softmax layer in language generation tasks, which results in faster and better generation quality [20]. Motivated by the success of these models, we propose to learn unsupervised text embeddings in the spherical space so that the embedding space discrepancy between training and usage can be eliminated, and directional similarity is more effectively captured.

## 3 Spherical Text Embedding

In this section, we introduce the spherical generative model for jointly learning word and paragraph embeddings and the corresponding loss function.

### 3.1 The Generative Model

The design of our generative model is inspired by the way humans write articles: Each word should be semantically consistent with not only its surrounding words, but also the entire paragraph/document.

Specifically, we assume text generation is a two-step process: A center word is first generated according to the semantics of the paragraph, and then the surrounding words are generated based on the center word's semantics[2]. Further, we assume the direction in the spherical embedding space captures textual semantics, and higher directional similarity implies higher co-occurrence probability. Hence, we model the text generation process as follows: Given a paragraph $d$, a center word $u$ is first generated by

$$p(u \mid d) \propto \exp(\cos(\boldsymbol{u}, \boldsymbol{d})), \tag{1}$$

and then a context word is generated by

$$p(v \mid u) \propto \exp(\cos(\boldsymbol{v}, \boldsymbol{u})), \tag{2}$$

where $\|\boldsymbol{u}\| = \|\boldsymbol{v}\| = \|\boldsymbol{d}\| = 1$, and $\cos(\cdot, \cdot)$ denotes the cosine of the angle between two vectors on the unit sphere.

Next we derive the analytic forms of Equations (1) and (2).

**Theorem 1.** When the corpus has infinite vocabulary, *i.e.*, $|V| \to \infty$, the analytic forms of Equations (1) and (2) are given by the von Mises-Fisher (vMF) distribution with the prior embedding as the mean direction and constant 1 as the concentration parameter, *i.e.*,

$$\lim_{|V| \to \infty} p(v \mid u) = \text{vMF}_p(\boldsymbol{v}; \boldsymbol{u}, 1), \quad \lim_{|V| \to \infty} p(u \mid d) = \text{vMF}_p(\boldsymbol{u}; \boldsymbol{d}, 1).$$

The proof of Theorem 1 can be found in Appendix A.

The vMF distribution defines a probability density over a hypersphere and is parameterized by a mean vector $\boldsymbol{\mu}$ and a concentration parameter $\kappa$. The probability density closer to $\boldsymbol{\mu}$ is greater and the spread is controlled by $\kappa$. Formally, A unit random vector $\boldsymbol{x} \in \mathbb{S}^{p-1}$ has the $p$-variate vMF distribution $\text{vMF}_p(\boldsymbol{x}; \boldsymbol{\mu}, \kappa)$ if its probability dense function is

$$f(\boldsymbol{x}; \boldsymbol{\mu}, \kappa) = c_p(\kappa) \exp\left(\kappa \cdot \cos(\boldsymbol{x}, \boldsymbol{\mu})\right),$$

where $\|\boldsymbol{\mu}\| = 1$ is the mean direction, $\kappa \geq 0$ is the concentration parameter, and the normalization constant $c_p(\kappa)$ is given by

$$c_p(\kappa) = \frac{\kappa^{p/2-1}}{(2\pi)^{p/2} I_{p/2-1}(\kappa)},$$

where $I_r(\cdot)$ represents the modified Bessel function of the first kind at order $r$, given by Definition 1 in the appendix.

Finally, the probability density function of a context word $v$ appearing in a center word $u$'s local context window in a paragraph/document $d$ is given by

$$p(v, u \mid d) \propto p(v \mid u) \cdot p(u \mid d) \propto \text{vMF}_p(\boldsymbol{v}; \boldsymbol{u}, 1) \cdot \text{vMF}_p(\boldsymbol{u}; \boldsymbol{d}, 1).$$

### 3.2 Objective

Given a positive training tuple $(u, v, d)$ where $v$ appears in the local context window of $u$ in paragraph $d$, we aim to maximize the probability $p(v, u \mid d)$, while minimize the probability $p(v, u' \mid d)$ where $u'$ is a randomly sampled word from the vocabulary serving as a negative sample. This is similar to the negative sampling technique used in Word2Vec [30] and GloVe [33]. To achieve this, we employ a max-margin loss function, similar to [15, 40, 41], and push the log likelihood of the positive tuple over the negative one by a margin:

$$\mathcal{L}(\boldsymbol{u}, \boldsymbol{v}, \boldsymbol{d}) = \max\left(0, m - \log\left(c_p(1)\exp(\cos(\boldsymbol{v}, \boldsymbol{u})) \cdot c_p(1)\exp(\cos(\boldsymbol{u}, \boldsymbol{d}))\right)\right.$$

$$\left. + \log\left(c_p(1)\exp(\cos(\boldsymbol{v}, \boldsymbol{u}')) \cdot c_p(1)\exp(\cos(\boldsymbol{u}', \boldsymbol{d}))\right)\right) \tag{3}$$

$$= \max\left(0, m - \cos(\boldsymbol{v}, \boldsymbol{u}) - \cos(\boldsymbol{u}, \boldsymbol{d}) + \cos(\boldsymbol{v}, \boldsymbol{u}') + \cos(\boldsymbol{u}', \boldsymbol{d})\right),$$

where $m > 0$ is the margin.

# 4 Optimization

In this section, we describe the approach to optimize the objective introduced in the previous section on the unit sphere.

## 4.1 The Constrained Optimization Problem

The unit hypersphere $\mathbb{S}^{p-1} := \{\boldsymbol{x} \in \mathbb{R}^p \mid \|\boldsymbol{x}\| = 1\}$ is the common choice for spherical space optimization problems. The text embedding training is thus a constrained optimization problem:

$$\min_{\boldsymbol{\Theta}} \mathcal{L}(\boldsymbol{\Theta}) \quad \text{s.t.} \quad \forall \boldsymbol{\theta} \in \boldsymbol{\Theta} : \|\boldsymbol{\theta}\| = 1,$$

where $\boldsymbol{\Theta} = \{\boldsymbol{u}_i\}\big|_{i=1}^{|V|} \cup \{\boldsymbol{v}_i\}\big|_{i=1}^{|V|} \cup \{\boldsymbol{d}_i\}\big|_{i=1}^{|\mathcal{D}|}$ is the set of target word embeddings, context word embeddings and paragraph embeddings to be learned.

Since the optimization problem is constrained on the unit sphere, the Euclidean space optimization methods such as SGD cannot be used to optimize our objective, because the Euclidean gradient provides the update direction in a non-curvature space, while the parameters in our model must be updated on a surface with constant positive curvature. Therefore, we need to base our embedding training problem on Riemannian optimization.

## 4.2 Preliminaries

A Riemannian manifold $(\mathcal{M}, g)$ is a real, smooth manifold whose tangent spaces are endowed with a smoothly varying inner product $g$, also called the Riemannian metric. Let $T_{\boldsymbol{x}}\mathcal{M}$ denote the tangent space at $\boldsymbol{x} \in \mathcal{M}$, then $g$ defines the inner product $\langle \cdot, \cdot \rangle_{\boldsymbol{x}} : T_{\boldsymbol{x}}\mathcal{M} \times T_{\boldsymbol{x}}\mathcal{M} \to \mathbb{R}$. A unit sphere $\mathbb{S}^{p-1}$ can be considered as a Riemmannian submanifold of $\mathbb{R}^p$, and its Riemannian metric can be inherited from $\mathbb{R}^p$, *i.e.*, $\langle \boldsymbol{\alpha}, \boldsymbol{\beta} \rangle_{\boldsymbol{x}} := \boldsymbol{\alpha}^\top \boldsymbol{\beta}$.

The intrinsic distance on the unit sphere between two arbitrary points $\boldsymbol{x}, \boldsymbol{y} \in \mathbb{S}^{p-1}$ is defined by $d(\boldsymbol{x}, \boldsymbol{y}) := \arccos(\boldsymbol{x}^\top \boldsymbol{y})$. A geodesic segment $\gamma : [a, b] \to \mathbb{S}^{p-1}$ is the generalization of a straight line to the sphere, and it is said to be minimal if it equals to the intrinsic distance between its end points, *i.e.*, $\ell(\gamma) = \arccos(\gamma(a)^\top \gamma(b))$.

Let $T_{\boldsymbol{x}}\mathbb{S}^{p-1}$ denote the tangent hyperplane at $\boldsymbol{x} \in \mathbb{S}^{p-1}$, *i.e.*, $T_{\boldsymbol{x}}\mathbb{S}^{p-1} := \{\boldsymbol{y} \in \mathbb{R}^p \mid \boldsymbol{x}^\top \boldsymbol{y} = 0\}$. The projection onto $T_{\boldsymbol{x}}\mathbb{S}^{p-1}$ is given by the linear mapping $I - \boldsymbol{x}\boldsymbol{x}^\top : \mathbb{R}^p \to T_{\boldsymbol{x}}\mathbb{S}^{p-1}$ where $I$ is the identity matrix. The exponential mapping $\exp_{\boldsymbol{x}} : T_{\boldsymbol{x}}\mathbb{S}^{p-1} \to \mathbb{S}^{p-1}$ projects a tangent vector $\boldsymbol{z} \in T_{\boldsymbol{x}}\mathbb{S}^{p-1}$ onto the sphere such that $\exp_{\boldsymbol{x}}(\boldsymbol{z}) = \boldsymbol{y}$, $\gamma(0) = \boldsymbol{x}$, $\gamma(1) = \boldsymbol{y}$ and $\frac{\partial}{\partial t}\gamma(0) = \boldsymbol{z}$.

## 4.3 Riemannian Optimization

Since the unit sphere is a Riemannian manifold, we can optimize our objectives with Riemannian SGD [6, 34]. Specifically, the parameters are updated by

$$\boldsymbol{x}_{t+1} = \exp_{\boldsymbol{x}_t}\left(-\eta_t \operatorname{grad} f(\boldsymbol{x}_t)\right),$$

where $\eta_t$ denotes the learning rate and $\operatorname{grad} f(\boldsymbol{x}_t) \in T_{\boldsymbol{x}_t}\mathbb{S}^{p-1}$ is the Riemannian gradient of a differentiable function $f : \mathbb{S}^{p-1} \to \mathbb{R}$.

On the unit sphere, the exponential mapping $\exp_{\boldsymbol{x}} : T_{\boldsymbol{x}}\mathbb{S}^{p-1} \to \mathbb{S}^{p-1}$ is given by

$$\exp_{\boldsymbol{x}}(\boldsymbol{z}) := \begin{cases} \cos(\|\boldsymbol{z}\|)\boldsymbol{x} + \sin(\|\boldsymbol{z}\|)\frac{\boldsymbol{z}}{\|\boldsymbol{z}\|}, & \boldsymbol{z} \in T_{\boldsymbol{x}}\mathbb{S}^{p-1} \backslash \{\boldsymbol{0}\}, \\ \boldsymbol{x}, & \boldsymbol{z} = \boldsymbol{0}. \end{cases} \tag{4}$$

To derive the Riemannian gradient $\operatorname{grad} f(\boldsymbol{x})$ at $\boldsymbol{x}$, we view $\mathbb{S}^{p-1}$ as a Riemannian submanifold of $\mathbb{R}^p$ endowed with the canonical Riemannian metric $\langle \boldsymbol{\alpha}, \boldsymbol{\beta} \rangle_{\boldsymbol{x}} := \boldsymbol{\alpha}^\top \boldsymbol{\beta}$. Then the Riemannian gradient is obtained by using the linear mapping $I - \boldsymbol{x}\boldsymbol{x}^\top : \mathbb{R}^p \to T_{\boldsymbol{x}}\mathbb{S}^{p-1}$ to project the Euclidean gradient $\nabla f(\boldsymbol{x})$ from the ambient Euclidean space onto the tangent hyperplane [1, 12], *i.e.*,

$$\operatorname{grad} f(\boldsymbol{x}) := \left(I - \boldsymbol{x}\boldsymbol{x}^\top\right) \nabla f(\boldsymbol{x}). \tag{5}$$

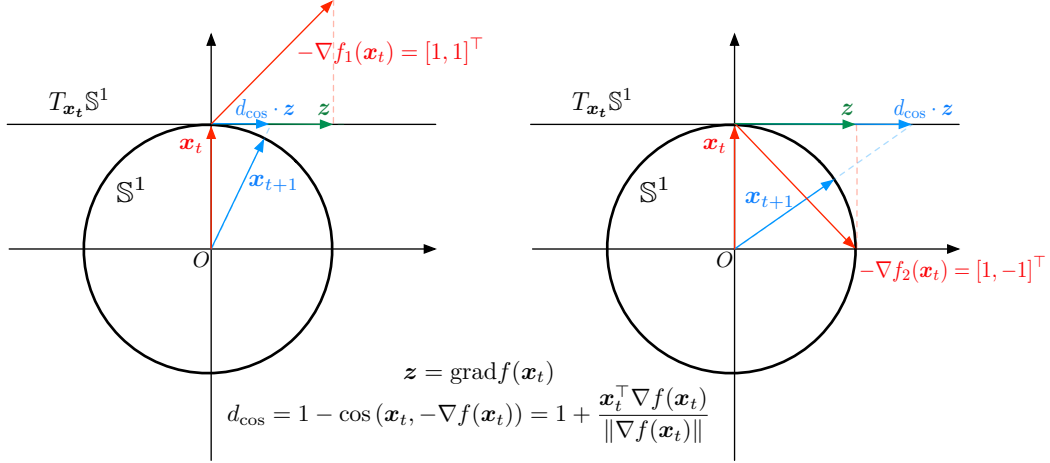

$$\boldsymbol{z} = \operatorname{grad} f(\boldsymbol{x}_t)$$
$$d_{\cos} = 1 - \cos\left(\boldsymbol{x}_t, -\nabla f(\boldsymbol{x}_t)\right) = 1 + \frac{\boldsymbol{x}_t^\top \nabla f(\boldsymbol{x}_t)}{\|\nabla f(\boldsymbol{x}_t)\|}$$

Figure 1: Example of modified Riemannian gradient descent on $\mathbb{S}^1$. Without modification, two Euclidean descent directions $-\nabla f_1(\boldsymbol{x}_t)$ and $-\nabla f_2(\boldsymbol{x}_t)$ give the same Riemannian gradient $\boldsymbol{z}$. We propose to multiply $\boldsymbol{z}$ with the cosine distance between $\boldsymbol{x}_t$ and $-\nabla f(\boldsymbol{x}_t)$ as the modified Riemannian gradient so that angular distances are taken into account during the parameter update.

## 4.4 Training Details

We describe two sets of design that lead to more efficient and effective training of the above optimization procedure.

First, the exponential mapping requires computation of non-linear functions, specifically, $\sin(\cdot)$ and $\cos(\cdot)$ in Equation (4), which is inefficient especially when the corpus is large. To tackle this issue, we can use a first-order approximation of the exponential mapping, called a retraction, *i.e.*, $R_{\boldsymbol{x}}(\boldsymbol{z}) : T_{\boldsymbol{x}}\mathbb{S}^{p-1} \to \mathbb{S}^{p-1}$ such that $d(R_{\boldsymbol{x}}(\boldsymbol{z}), \exp_{\boldsymbol{x}}(\boldsymbol{z})) = O(\|\boldsymbol{z}\|^2)$. For the unit sphere, we can simply define the retraction $R_{\boldsymbol{x}}(\boldsymbol{z})$ to be an addition in the ambient Euclidean space followed by a projection onto the sphere [1, 6], *i.e.*,

$$R_{\boldsymbol{x}}(\boldsymbol{z}) \coloneqq \frac{\boldsymbol{x} + \boldsymbol{z}}{\|\boldsymbol{x} + \boldsymbol{z}\|}. \tag{6}$$

Second, the Riemannian gradient given by Equation (5) provides the correct direction to update the parameters on the sphere, but its norm is not optimal when our goal is to train embeddings that capture directional similarity. This issue can be illustrated by the following toy example shown in Figure 1: Consider a point $\boldsymbol{x}_t$ with Euclidean coordinate $(0, 1)$ on a 2-d unit sphere $\mathbb{S}^1$ and two Euclidean gradient descent directions $-\nabla f_1(\boldsymbol{x}_t) = [1, 1]^\top$ and $-\nabla f_2(\boldsymbol{x}_t) = [1, -1]^\top$. Also, for simplicity, assume $\eta_t = 1$. In this case, the Riemannian gradient projected from $-\nabla f_2(\boldsymbol{x}_t)$ is the same with that of $-\nabla f_1(\boldsymbol{x}_t)$ and is equal to $[1, 0]^\top$, *i.e.*, $\operatorname{grad} f_1(\boldsymbol{x}_t) = \operatorname{grad} f_2(\boldsymbol{x}_t) = [1, 0]^\top$. However, when our goal is to capture directional information and distance is measured by the angles between vectors, $-\nabla f_2(\boldsymbol{x}_t)$ suggests a bigger step to take than $-\nabla f_1(\boldsymbol{x}_t)$ at point $\boldsymbol{x}_t$. To explicitly incorporate angular distance into the optimization procedure, we use the cosine distance between the current point $\boldsymbol{x}_t \in \mathbb{S}^{p-1}$ and the Euclidean gradient descent direction $-\nabla f(\boldsymbol{x}_t)$, *i.e.*, $\left(1 + \frac{\boldsymbol{x}_t^\top \nabla f(\boldsymbol{x}_t)}{\|\nabla f(\boldsymbol{x}_t)\|}\right)$, as a multiplier to the computed Riemannian gradient according to Equation (5). The rationale of this design is to encourage parameters with greater cosine distance from its target direction to take a larger update step. We find that when updating negative samples, it is empirically better to use negative cosine similarity instead of cosine distance as the multiplier to the Riemannian gradient. This is probably because negative samples are randomly sampled from the vocabulary and most of them are semantically irrelevant with the center word. Therefore, their ideal embeddings should be orthogonal to the center word's embedding. However, using cosine distance will encourage them to point to the opposite direction of the center word's embedding.

In summary, with the above designs, we optimize the parameters by the following update rule:

$$\boldsymbol{x}_{t+1} = R_{\boldsymbol{x}_t}\left(-\eta_t \left(1 + \frac{\boldsymbol{x}_t^\top \nabla f(\boldsymbol{x}_t)}{\|\nabla f(\boldsymbol{x}_t)\|}\right)\left(I - \boldsymbol{x}_t \boldsymbol{x}_t^\top\right) \nabla f(\boldsymbol{x}_t)\right). \tag{7}$$

Finally, we provide the convergence guarantee of the above update rule when applied to optimize our objectives.

**Theorem 2.** When the update rule given by Equation (7) is applied to $\mathcal{L}(\boldsymbol{x})$, and the learning rate satisfies the usual condition in stochastic approximation, *i.e.*, $\sum_t \eta_t^2 < \infty$ and $\sum_t \eta_t = \infty$, $\boldsymbol{x}$ converges almost surely to a critical point $\boldsymbol{x}^*$ and grad $\mathcal{L}(\boldsymbol{x})$ converges almost surely to 0, *i.e.*,

$$\Pr\left(\lim_{t\to\infty}\mathcal{L}(\boldsymbol{x}_t) = \mathcal{L}(\boldsymbol{x}^*)\right) = 1, \quad \Pr\left(\lim_{t\to\infty}\operatorname{grad}\mathcal{L}(\boldsymbol{x}_t) = 0\right) = 1.$$

The proof of Theorem 2 can be found in Appendix B.

## 5 Evaluation

In this section, we empirically evaluate the quality of spherical text embeddings for three common text embedding application tasks, *i.e.*, word similarity, document clustering and document classification. Our model is named **JoSE**, for **Jo**int **S**pherical **E**mbedding. For all three tasks, our spherical embeddings and all baselines are trained according to the following setting: The models are trained for 10 iterations on the corpus; the local context window size is 10; the embedding dimension is 100. In our **JoSE** model, we set the margin in Equation (3) to be 0.15, the number of negative samples to be 2, the initial learning rate to be 0.04 with linear decay. Other hyperparameters are set to be the default value of the corresponding algorithm.

Also, since text embeddings serve as the building block for many downstream tasks, it is essential that the embedding training is efficient and can scale to very large datasets. At the end of this section, we will provide the training time of different word embedding models when trained on the latest Wikipedia dump.

### 5.1 Word Similarity

We conduct word similarity evaluation on the following benchmark datasets: WordSim353 [13], MEN [7] and SimLex999 [17]. The training corpus for word similarity is the latest Wikipedia dump[3] containing 2.4 billion tokens. Words appearing less than 100 times are discarded, leaving 239, 672 unique tokens. The Spearman's rank correlation is reported in Table 1, which reflects the consistency between word similarity rankings given by cosine similarity of word embeddings and human raters. We compare our model with the following baselines: Word2Vec [30], GloVe [33], fastText [5] and BERT [10] which are trained in Euclidean space, and Poincaré GloVe [39] which is trained in Poincaré space. The results demonstrate that training embeddings in the spherical space is essential for the superior performance on word similarity. We attempt to explain why the recent popular language model, BERT [10], falls behind other baselines on this task: (1) BERT learns contextualized representations, but word similarity evaluation is conducted in a context-free manner; averaging contextualized representations to derive context-free representations may not be the intended usage of BERT. (2) BERT is optimized on specific downstream tasks like predicting masked words and sentence relationships, which have no direct relation to word similarity.

Table 1: Spearman rank correlation on word similarity evaluation.

| Embedding Space | Model | WordSim353 | MEN | SimLex999 |
|---|---|---|---|---|
| Euclidean | Word2Vec | 0.711 | 0.726 | 0.311 |
| | GloVe | 0.598 | 0.710 | 0.321 |
| | fastText | 0.697 | 0.722 | 0.303 |
| | BERT | 0.477 | 0.594 | 0.287 |
| Poincaré | Poincaré GloVe | 0.623 | 0.652 | 0.321 |
| Spherical | **JoSE** | **0.739** | **0.748** | **0.339** |

## 5.2 Document Clustering

We perform document clustering to evaluate the quality of the spherical paragraph embeddings trained by our model. The training corpus is the 20 Newsgroups dataset[4], and we treat each document as a paragraph in all compared models. The dataset contains around $18,000$ newsgroup documents (both training and testing documents are used) partitioned into 20 classes. For clustering methods, we use both K-Means and spherical K-Means (SK-Means) [3] which performs clustering in the spherical space. We compare with the following paragraph embedding baselines: Averaged word embedding using Word2Vec [30], SIF [2], BERT [10] and Doc2Vec [22]. We use four widely used external measures [3, 25, 36] as metrics: Mutual Information (MI), Normalized Mutual Information (NMI), Adjusted Rand Index (ARI), and Purity. The results are reported in Table 2, with mean and standard deviation computed over 10 runs. It is shown that feature quality is generally more important that clustering algorithms for document clustering tasks: Using spherical K-Means only gives marginal performance boost over K-Means, while **JoSE** remains optimal regardless of clustering algorithms. This demonstrates that directional similarity on document/paragraph-level features is beneficial also for clustering tasks, which can be captured intrinsically in the spherical space.

Table 2: Document clustering evaluation on the 20 Newsgroup dataset.

| Embedding | Clus. Alg. | MI | NMI | ARI | Purity |
|---|---|---|---|---|---|
| Avg. W2V | K-Means | $1.299 \pm 0.031$ | $0.445 \pm 0.009$ | $0.247 \pm 0.008$ | $0.408 \pm 0.014$ |
| | SK-Means | $1.328 \pm 0.024$ | $0.453 \pm 0.009$ | $0.250 \pm 0.008$ | $0.419 \pm 0.012$ |
| SIF | K-Means | $0.893 \pm 0.028$ | $0.308 \pm 0.009$ | $0.137 \pm 0.006$ | $0.285 \pm 0.011$ |
| | SK-Means | $0.958 \pm 0.012$ | $0.322 \pm 0.004$ | $0.164 \pm 0.004$ | $0.331 \pm 0.005$ |
| BERT | K-Means | $0.719 \pm 0.013$ | $0.248 \pm 0.004$ | $0.100 \pm 0.003$ | $0.233 \pm 0.005$ |
| | SK-Means | $0.854 \pm 0.022$ | $0.289 \pm 0.008$ | $0.127 \pm 0.003$ | $0.281 \pm 0.010$ |
| Doc2Vec | K-Means | $1.856 \pm 0.020$ | $0.626 \pm 0.006$ | $0.469 \pm 0.015$ | $0.640 \pm 0.016$ |
| | SK-Means | $1.876 \pm 0.020$ | $0.630 \pm 0.007$ | $0.494 \pm 0.012$ | $0.648 \pm 0.017$ |
| **JoSE** | K-Means | $1.975 \pm 0.026$ | $0.663 \pm 0.008$ | $0.556 \pm 0.018$ | $0.711 \pm 0.020$ |
| | SK-Means | $\mathbf{1.982} \pm 0.034$ | $\mathbf{0.664} \pm 0.010$ | $\mathbf{0.568} \pm 0.020$ | $\mathbf{0.721} \pm 0.029$ |

## 5.3 Document Classification

Apart from document clustering, we also evaluate the quality of spherical paragraph embeddings on document classification tasks. Besides the 20 Newsgroup dataset used in Section 5.2 which is a topic classification dataset, we evaluate different document/paragraph embedding methods also on a binary sentiment classification dataset consisting of $1,000$ positive and $1,000$ negative movie reviews[5]. We again treat each document in both datasets as a paragraph in all models. For the 20 Newsgroup dataset, we follow the original train/test sets split; for the movie review dataset, we randomly select $80\%$ of the data as training and $20\%$ as testing. We use $k$-NN [9] as the classification algorithm with Euclidean distance as the distance metric. Since $k$-NN is a non-parametric method, the performances of $k$-NN directly reflect how well the topology of the embedding space captures document-level semantics (*i.e.*, whether documents from the same semantic class are embedded closer). We set $k = 3$ in the experiment (we observe similar comparison results when ranging $k$ in $[1, 10]$) and report the performances of all methods measured by Macro-F1 and Micro-F1 scores in Table 3. **JoSE** achieves the best performances on both datasets with $k$-NN classification, demonstrating the effectiveness of **JoSE** in capturing both topical and sentiment semantics into learned paragraph embeddings.

## 5.4 Training Efficiency

We report the training time on the latest Wikipedia dump per iteration of all baselines used in Section 5.1 to compare the training efficiency. All the models except BERT are run on a machine with 20 cores of Intel(R) Xeon(R) CPU E5-2680 v2 @ 2.80 GHz; BERT is trained on $8$ NVIDIA

Table 3: Document classification evaluation using $k$-NN ($k = 3$).

| Embedding | 20 Newsgroup | | Movie Review | |
|---|---|---|---|---|
| | Macro-F1 | Micro-F1 | Macro-F1 | Micro-F1 |
| Avg. W2V | 0.630 | 0.631 | 0.712 | 0.713 |
| SIF | 0.552 | 0.549 | 0.650 | 0.656 |
| BERT | 0.380 | 0.371 | 0.664 | 0.665 |
| Doc2Vec | 0.648 | 0.645 | 0.674 | 0.678 |
| **JoSE** | **0.703** | **0.707** | **0.764** | **0.765** |

GeForce GTX 1080 GPUs. The training time is reported in Table 4. All text embedding frameworks are able to scale to large datasets (except BERT which is not specifically designed for learning text embeddings), but **JoSE** enjoys the highest efficiency. The overall efficiency of our model results from both our objective function design and the optimization procedure: (1) The objective of our model (Equation (3)) only contains simple operations (note that cosine similarity on the unit sphere is simply vector dot product), while other models contains non-linear operations (Word2Vec's and fastText's objectives involve exponential functions; GloVe's objective involves logarithm functions); (2) After replacing the original exponential mapping (Equation (4)) with retraction (Equation (6)), the update rule (Equation (7)) only computes vector additions, multiplications and normalization in addition to the Euclidean gradient, which are all inexpensive operations.

Table 4: Training time (per iteration) on the latest Wikipedia dump.

| Word2Vec | GloVe | fastText | BERT | Poincaré GloVe | **JoSE** |
|---|---|---|---|---|---|
| 0.81 hrs | 0.85 hrs | 2.11 hrs | > 5 days | 1.25 hrs | **0.73 hrs** |

# 6   Conclusions and Future Work

In this paper, we propose to address the discrepancy between the training procedure and the practical usage of Euclidean text embeddings by learning spherical text embeddings that intrinsically captures directional similarity. Specifically, we introduce a spherical generative model consisting of a two-step generative process to jointly learn word and paragraph embeddings. Furthermore, we develop an efficient Riemannian optimization method to train text embeddings on the unit hypersphere. State-of-the-art results on common text embedding applications including word similarity and document clustering demonstrate the efficacy of our model. With a simple training objective and an efficient optimization procedure, our proposed model enjoys better efficiency compared to previous embedding learning systems.

In future work, it will be interesting to exploit spherical embedding space for other tasks like lexical entailment, by also learning the concentration parameter in the vMF distribution of each word in the generative model or designing other generative models. It may also be possible to incorporate other signals such as subword information [5] into spherical text embeddings learning for even better embedding quality. Our unsupervised embedding model may also benefit other supervised tasks: Since word embeddings are commonly used as the first layer in deep neural networks, it might be beneficial to either add norm constraints or apply Riemannian optimization when fine-tuning the word embedding layer.

### Acknowledgments

Research was sponsored in part by U.S. Army Research Lab. under Cooperative Agreement No. W911NF-09-2-0053 (NSCTA), DARPA under Agreements No. W911NF-17-C-0099 and FA8750-19-2-1004, National Science Foundation IIS 16-18481, IIS 17-04532, and IIS 17-41317, DTRA HDTRA11810026, and grant 1U54GM114838 awarded by NIGMS through funds provided by the trans-NIH Big Data to Knowledge (BD2K) initiative (www.bd2k.nih.gov). Any opinions, findings, and conclusions or recommendations expressed in this document are those of the author(s) and should not be interpreted as the views of any U.S. Government. The U.S. Government is authorized to

reproduce and distribute reprints for Government purposes notwithstanding any copyright notation hereon. We thank anonymous reviewers for valuable and insightful feedback.

## Footnotes

*Currently at Google Research.

[2]Like previous works, we assume each word has independent center word representation and context word representation, and thus the generation processes of a word as a center word and as a context word are independent.

[3]`https://dumps.wikimedia.org/enwiki/latest/enwiki-latest-pages-articles.xml.bz2`

[4] http://qwone.com/~jason/20Newsgroups/

[5] http://www.cs.cornell.edu/people/pabo/movie-review-data/

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
