[Supplementary Material]

# Supplementary Material: Spherical Text Embedding

## A    Proof of Theorem 1

**Definition 1** (Modified Bessel Function of the First Kind). The modified Bessel function of the first kind of order $r$ can be defined as [26]:

$$I_r(\kappa) = \frac{(\kappa/2)^r}{\Gamma\left(r + \frac{1}{2}\right)\Gamma\left(\frac{1}{2}\right)} \int_0^\pi \exp(\kappa\cos\theta)(\sin\theta)^{2r}d\theta,$$

where $\Gamma(x) = \int_0^\infty \exp(-t)t^{x-1}dt$ is the gamma function.

**Lemma 1.** The definite integral of power of $\sin$ on the interval $[0, \pi]$ is given by

$$J_p = \int_0^\pi (\sin x)^p dx = \frac{\sqrt{\pi}\Gamma\left(\frac{1+p}{2}\right)}{\Gamma\left(1 + \frac{p}{2}\right)} \quad (p \in \mathbb{Z}^+),$$

where $\Gamma(x) = \int_0^\infty \exp(-t)t^{x-1}dt$ is the gamma function.

*Proof.*

$$J_p = \int_0^\pi (\sin x)^p dx$$

$$= \left(-(\sin x)^{p-1}\cos x\right)\Big|_0^\pi + (p-1)\int_0^\pi (\sin x)^{p-2}(\cos x)^2 dx$$

$$= 0 + (p-1)\int_0^\pi (\sin x)^{p-2}dx - (p-1)\int_0^\pi (\sin x)^p dx$$

$$= (p-1)J_{p-2} - (p-1)J_p$$

Therefore, $J_p = \frac{p-1}{p}J_{p-2}$.

Using the above iteration relationship and the property of gamma function $\Gamma(x+1) = x\Gamma(x)$, we write $J_p$ using gamma function:

- When $p$ is an even integer:

$$J_p = \frac{p-1}{p}\frac{p-3}{p-2}\cdots\frac{1}{2}J_0$$

$$= \frac{(p-1)/2}{p/2}\frac{(p-3)/2}{(p-2)/2}\cdots\frac{1/2}{2/2}J_0$$

$$= \frac{\Gamma\left(\frac{1+p}{2}\right)/\Gamma\left(\frac{1}{2}\right)}{\Gamma\left(\frac{p}{2}+1\right)/\Gamma(1)}J_0$$

Plugging in the base case $J_0 = \pi$ and $\Gamma\left(\frac{1}{2}\right) = \sqrt{\pi}$, $\Gamma(1) = 1$, we prove that

$$J_p = \frac{\sqrt{\pi}\Gamma\left(\frac{1+p}{2}\right)}{\Gamma\left(1 + \frac{p}{2}\right)} \quad (p \in \mathbb{Z}^+, p \text{ is even})$$

- When $p$ is an odd integer:

$$J_p = \frac{p-1}{p}\frac{p-3}{p-2}\cdots\frac{2}{3}J_1$$

$$= \frac{(p-1)/2}{p/2}\frac{(p-3)/2}{(p-2)/2}\cdots\frac{2/2}{3/2}J_1$$

$$= \frac{\Gamma\left(\frac{1+p}{2}\right)/\Gamma(1)}{\Gamma\left(\frac{p}{2}+1\right)/\Gamma\left(\frac{3}{2}\right)}J_1$$

Plugging in the base case $J_1 = 2$ and $\Gamma\left(\frac{3}{2}\right) = \frac{\sqrt{\pi}}{2}$, $\Gamma(1) = 1$, we prove that

$$J_p = \frac{\sqrt{\pi}\Gamma\left(\frac{1+p}{2}\right)}{\Gamma\left(1 + \frac{p}{2}\right)} \quad (p \in \mathbb{Z}^+, p \text{ is odd})$$

$\square$

**Theorem 1.** When the corpus has infinite vocabulary, *i.e.*, $|V| \to \infty$, the analytic forms of Equations (1) and (2) are given by the von Mises-Fisher (vMF) distribution with the prior embedding as the mean direction and constant 1 as the concentration parameter, *i.e.*,

$$\lim_{|V| \to \infty} p(v \mid u) = \text{vMF}_p(\boldsymbol{v}; \boldsymbol{u}, 1), \quad \lim_{|V| \to \infty} p(u \mid d) = \text{vMF}_p(\boldsymbol{u}; \boldsymbol{d}, 1).$$

*Proof.* We give the proof for the first equality, and the second equality can be derived similarly. We generalize the relationship proportionality $p(\boldsymbol{v} \mid \boldsymbol{u}) \propto \exp(\exp(\cos(\boldsymbol{v}, \boldsymbol{u})))$ in Equation (2) to the continuous case and obtain the following probability dense distribution:

$$\lim_{|V| \to \infty} p(v \mid u) = \frac{\exp(\cos(\boldsymbol{v}, \boldsymbol{u}))}{\int_{\mathbb{S}^{p-1}} \exp(\cos(\boldsymbol{v}', \boldsymbol{u})) d\boldsymbol{v}'} \triangleq \frac{\exp(\cos(\boldsymbol{v}, \boldsymbol{u})))}{Z}, \tag{8}$$

where we denote the integral in the denominator as $Z$.

To evaluate the integral $Z$, we make the transformation to polar coordinates. Let $\boldsymbol{t} = Q\boldsymbol{v}'$, where $Q \in \mathbb{R}^{p \times p}$ is an orthogonal transformation so that $d\boldsymbol{t} = d\boldsymbol{v}'$. Moreover, let the first row of $Q$ be $\boldsymbol{u}$ so that $t_1 = \boldsymbol{u}^\top \boldsymbol{v}'$. Then we use $(r, \theta_1, \ldots, \theta_{p-1})$ to represent the polar coordinates of $\boldsymbol{t}$ where $r = 1$ and $\cos\theta_1 = \boldsymbol{u}^\top \boldsymbol{v}$. The transformation from Euclidean coordinates to polar coordinates is given by [35] via computing the determinant of the Jacobian matrix for the coordinate transformation:

$$d\boldsymbol{t} = r^{p-1} \prod_{j=2}^{p} (\sin\theta_{j-1})^{p-j} d\theta_{j-1}.$$

Then

$$Z = \int_0^\pi \exp(\cos\theta_1)(\sin\theta_1)^{p-2} d\theta_1 \prod_{j=3}^{p-1} \int_0^\pi (\sin\theta_{j-1})^{p-j} d\theta_{j-1} \int_0^{2\pi} d\theta_{j-1}.$$

By Lemma 1, we have

$$\prod_{j=3}^{p-1} \int_0^\pi (\sin\theta_{j-1})^{p-j} d\theta_{j-1} = \pi^{\frac{p-3}{2}} \frac{\Gamma\left(\frac{p-2}{2}\right)\Gamma\left(\frac{p-3}{2}\right) \cdots \Gamma(1)}{\Gamma\left(\frac{p-1}{2}\right)\Gamma\left(\frac{p-2}{2}\right) \cdots \Gamma\left(\frac{3}{2}\right)} = \frac{\pi^{\frac{p-3}{2}}}{\Gamma\left(\frac{p-1}{2}\right)}.$$

Then

$$Z = \int_0^\pi \exp(\cos\theta_1)(\sin\theta_1)^{p-2} d\theta_1 \cdot \frac{\pi^{\frac{p-3}{2}}}{\Gamma\left(\frac{p-1}{2}\right)} \cdot 2\pi$$

$$= \frac{2\pi^{\frac{p-1}{2}}}{\Gamma\left(\frac{p-1}{2}\right)} \int_0^\pi \exp(\cos\theta_1)(\sin\theta_1)^{p-2} d\theta_1.$$

According to Definition 1, the integral term of $Z$ above can be expressed with $I_{p/2-1}(1)$ as:

$$\int_0^\pi \exp(\cos\theta_1)(\sin\theta_1)^{p-2} d\theta_1 = \frac{\Gamma\left(\frac{p-1}{2}\right)\Gamma\left(\frac{1}{2}\right)}{2^{1-p/2}} I_{p/2-1}(1).$$

Therefore, with the fact that $\Gamma\left(\frac{1}{2}\right) = \sqrt{\pi}$,

$$Z = \frac{2\pi^{\frac{p-1}{2}}}{\Gamma\left(\frac{p-1}{2}\right)} \frac{\Gamma\left(\frac{p-1}{2}\right)\Gamma\left(\frac{1}{2}\right)}{2^{1-p/2}} I_{p/2-1}(1) = (2\pi)^{p/2} I_{p/2-1}(1).$$

Plugging $Z$ back to Equation (8), we finally arrive that

$$\lim_{|V| \to \infty} p(v \mid u) = \frac{1}{(2\pi)^{p/2} I_{p/2-1}(1)} \exp(\cos(\boldsymbol{v}, \boldsymbol{u})) = \text{vMF}_p(\boldsymbol{v}; \boldsymbol{u}, 1).$$

$\square$

# B Proof of Theorem 2

**Lemma 2.** Let $(X_n)_{n\in\mathbb{N}}$ be a non-negative stochastic process with bounded positive variations, *i.e.*, $\sum_{n=0}^{\infty} \mathbb{E}\left[\max\left(\mathbb{E}\left[X_{n+1} - X_n \mid \mathcal{F}_n\right], 0\right)\right] < \infty$. Then this process is a quasi-martingale, *i.e.*,

$$\Pr\left(\sum_{n=0}^{\infty} \left|\mathbb{E}\left[X_{n+1} - X_n \mid \mathcal{F}_n\right]\right| < \infty\right) = 1, \quad \Pr\left(\lim_{n\to\infty} X_n = X^*\right) = 1,$$

where $\mathcal{F}_n$ is the increasing sequence of $\sigma$-algebras generated by variables before time $n$, $\mathcal{F}_n = \{s_0, \ldots, s_{n-1}\}$, such that $X_n$ computed from $s_0, \ldots, s_{n-1}$ is $\mathcal{F}_n$ measurable.

*Proof.* See [14]. $\qquad\qquad\qquad\qquad\qquad\qquad\qquad\qquad\qquad\qquad\qquad\qquad\qquad\qquad\square$

Before proving Theorem 2, we first prove the update rule without approximation, *i.e.*, replacing the retraction $R_{\boldsymbol{x}}$ in Equation (7) with the exponential map $\exp_{\boldsymbol{x}}$ defined by Equation (4), leads to almost surely convergence.

**Lemma 3.** When the update rule given by

$$\boldsymbol{x}_{t+1} = R_{\boldsymbol{x}_t}\left(-\eta_t\left(1 + \frac{\boldsymbol{x}_t^\top \nabla f(\boldsymbol{x}_t)}{\|\nabla f(\boldsymbol{x}_t)\|}\right)\left(I - \boldsymbol{x}_t \boldsymbol{x}_t^\top\right)\nabla f(\boldsymbol{x}_t)\right).$$

is applied to $\mathcal{L}(\boldsymbol{\theta})$, and the learning rate satisfies the usual condition in stochastic approximation, *i.e.*, $\sum_t \eta_t^2 < \infty$ and $\sum_t \eta_t = \infty$, $\boldsymbol{\theta}$ converges almost surely to a critical point $\boldsymbol{\theta}^*$ and $\operatorname{grad}\mathcal{L}(\boldsymbol{\theta})$ converges almost surely to $0$, *i.e.*,

$$\Pr\left(\lim_{t\to\infty} \mathcal{L}(\boldsymbol{\theta}_t) = \mathcal{L}(\boldsymbol{\theta}^*)\right) = 1, \quad \Pr\left(\lim_{t\to\infty} \operatorname{grad}\mathcal{L}(\boldsymbol{\theta}_t) = 0\right) = 1.$$

*Proof.* We use $l(\boldsymbol{x}_t, z_t)$ to denote the approximated loss function $\mathcal{L}(\boldsymbol{x}_t)$ evaluated at a training instance $z_t$, *i.e.*, $\mathcal{L}(\boldsymbol{x}_t) = \mathbb{E}_z[l(\boldsymbol{x}_t, z)]$.

Let

$$\xi(\boldsymbol{x}_t, z_t) = \eta_t\left(1 + \frac{\boldsymbol{x}_t^\top \nabla l(\boldsymbol{x}_t, z_t)}{\|\nabla l(\boldsymbol{x}_t, z_t)\|}\right), \quad \operatorname{grad} l(\boldsymbol{x}_t, z_t) = \left(I - \boldsymbol{x}_t \boldsymbol{x}_t^\top\right)\nabla l(\boldsymbol{x}_t, z_t),$$

and we omit the arguments of $\xi$ and $\operatorname{grad} l$ in the following derivation because we only care about the upper bound of both.

We consider two consecutive gradient update steps of the parameter, $\boldsymbol{x}_t$ and $\boldsymbol{x}_{t+1}$. There exists a geodesic segment $\gamma(s) = \exp_{\boldsymbol{x}_t}(-s \cdot \operatorname{grad} l)$ linking $\boldsymbol{x}_t$ and $\boldsymbol{x}_{t+1}$, and its length is obtained by

$$\ell(\gamma) = \int_0^\xi \|\gamma'(s)\|\, ds.$$

Then we apply the Taylor formula on Riemannian manifold [6] and have

$$\begin{aligned}
\mathcal{L}(\boldsymbol{x}_{t+1}) &= \mathcal{L}(\exp_{\boldsymbol{x}_t}(-\xi \cdot \operatorname{grad} l)) \\
&= \mathcal{L}(\boldsymbol{x}_t) - \xi \cdot \operatorname{grad} l^\top \operatorname{grad}\mathcal{L}(\boldsymbol{x}_t) + \int_0^\xi (\xi - s)\gamma'(s)^\top \operatorname{Hess}\mathcal{L}(\gamma(s))\gamma'(s)ds,
\end{aligned} \tag{9}$$

where $\operatorname{grad}\mathcal{L}$ is the Riemannian gradient satisfying

$$\left.\frac{d}{dt}\mathcal{L}(\exp_{\boldsymbol{x}}(t\boldsymbol{\alpha}))\right|_{t=0} = \boldsymbol{\alpha}^\top \operatorname{grad}\mathcal{L}(\boldsymbol{x}),$$

and $\operatorname{Hess}\mathcal{L}(\boldsymbol{x})$ is the Riemannian Hessian satisfying

$$\left.\frac{d}{dt}\left(\operatorname{grad}\mathcal{L}(\exp_{\boldsymbol{x}}(t\boldsymbol{\alpha}))^\top \operatorname{grad}\mathcal{L}(\exp_{\boldsymbol{x}}(t\boldsymbol{\alpha}))\right)\right|_{t=0} = 2\operatorname{grad}\mathcal{L}(\boldsymbol{x})^\top \operatorname{Hess}\mathcal{L}(\boldsymbol{x})\boldsymbol{\alpha}.$$

For spherical space, the Riemannian Hessian is given by [1, 12]

$$\operatorname{Hess} f(\boldsymbol{x}) := \left(I - \boldsymbol{x}\boldsymbol{x}^\top\right)\left(\nabla^2 f(\boldsymbol{x}) - \nabla f(\boldsymbol{x})^\top \boldsymbol{x} I\right),$$

where $\nabla^2 f(\boldsymbol{x})$ is the Euclidean Hessian of $f(\boldsymbol{x})$.

Therefore, at each training instance $z_t$, the Riemannian Hessian of the approximated loss is bounded:

$$
\begin{aligned}
\|\operatorname{Hess} l(\boldsymbol{x}_t, z_t)\| &\leq \left\|I - \boldsymbol{x}_t \boldsymbol{x}_t^\top\right\| \left\|\nabla^2 l(\boldsymbol{x}_t, z_t) - \nabla l(\boldsymbol{x}_t, z_t)^\top \boldsymbol{x}_t I\right\| \\
&\leq \left(1 + \|\boldsymbol{x}_t\|^2\right) \|\nabla l(\boldsymbol{x}_t, z_t)\| \|\boldsymbol{x}_t\| \\
&\leq 4,
\end{aligned}
$$

where we use the fact that $\nabla^2 l(\boldsymbol{x}_t, z_t) = 0$ and $\|\nabla l(\boldsymbol{x}_t, z_t)\| \leq 2$ by computing the Euclidean Hessian and gradient of Equation (3).

Consequently, the Riemannian Hessian of the original loss is bounded:

$$
\|\operatorname{Hess} \mathcal{L}(\boldsymbol{x}_t)\| \leq \mathbb{E}_z\left[\|\operatorname{Hess} l(\boldsymbol{x}_t, z)\|\right] \leq 4.
$$

Similarly, the Riemannian gradient is also bounded:

$$
\|\operatorname{grad} l(\boldsymbol{x}_t, z_t)\| \leq \left\|I - \boldsymbol{x}_t \boldsymbol{x}_t^\top\right\| \|\nabla l(\boldsymbol{x}_t, z_t)\| \leq 4.
$$

Also, $\xi$ is bounded by

$$
\xi(\boldsymbol{x}_t, z_t) = \eta_t \left(1 + \frac{\boldsymbol{x}_t^\top \nabla l(\boldsymbol{x}_t, z_t)}{\|\nabla l(\boldsymbol{x}_t, z_t)\|}\right) \leq 2\eta_t.
$$

Then the integral in Equation (9) is bounded by

$$
\int_0^\xi (\xi - s) \gamma'(s)^\top \operatorname{Hess} \mathcal{L}(\gamma(s)) \gamma'(s) ds \leq \xi^2 \|\operatorname{grad} l\|^2 \|\operatorname{Hess} \mathcal{L}(\boldsymbol{x}_t)\| \leq 256\eta_t^2.
$$

Therefore, Equation (9) shows that

$$
\mathcal{L}(\boldsymbol{x}_{t+1}) - \mathcal{L}(\boldsymbol{x}_t) \leq -\xi \cdot \operatorname{grad} l^\top \operatorname{grad} \mathcal{L}(\boldsymbol{x}_t) + 256\eta_t^2. \tag{10}
$$

Now let $\mathcal{F}_t$ be the increasing sequence of $\sigma$-algebras generated by variables before time $t$, $\mathcal{F}_t = \{z_0, \ldots, z_{t-1}\}$, such that $\boldsymbol{x}_t$ computed from $z_0, \ldots, z_{t-1}$ is $\mathcal{F}_t$ measurable.

Then we take the expectation over $z$ under $\mathcal{F}_t$ of both sides of Equation (10)

$$
\begin{aligned}
\mathbb{E}_z[\mathcal{L}(\boldsymbol{x}_{t+1}) - \mathcal{L}(\boldsymbol{x}_t) \mid \mathcal{F}_t] &\leq -\xi \mathbb{E}_z[\operatorname{grad} l(\boldsymbol{x}_t, z_t)^\top \operatorname{grad} \mathcal{L}(\boldsymbol{x}_t) \mid \mathcal{F}_t] + 256\eta_t^2 \\
&= -\xi \mathbb{E}_z[\operatorname{grad} l(\boldsymbol{x}_t, z_t)^\top \operatorname{grad} \mathcal{L}(\boldsymbol{x}_t)] + 256\eta_t^2 \\
&= -\xi \|\operatorname{grad} \mathcal{L}(\boldsymbol{x}_t)\|^2 + 256\eta_t^2,
\end{aligned} \tag{11}
$$

since $z_t$ is independent of $\mathcal{F}_t$.

As $\mathcal{L}(\boldsymbol{x}_t) \geq 0$ and $\sum_t \eta_t^2 < \infty$, this shows that $\mathcal{L}(\boldsymbol{x}_t) + \sum_t^\infty 256\eta_t^2$ is a non-negative supermartingale, thus it converges almost surely. Therefore, $\mathcal{L}(\boldsymbol{x}_t)$ converges almost surely.

Next, to prove $\operatorname{grad} \mathcal{L}(\boldsymbol{x}_t)$ converges, we repeat the above proof and replace $\mathcal{L}(\boldsymbol{x}_t)$ with $\|\operatorname{grad} \mathcal{L}(\boldsymbol{x}_t)\|^2$. Specifically, we can bound the second derivative of $\|\operatorname{grad} \mathcal{L}(\boldsymbol{x}_t)\|^2$ and arrive at a very similar form as Equation (11). We then prove $\operatorname{grad} \mathcal{L}(\boldsymbol{x}_t)$ almost surely converge.

Finally, we prove $\operatorname{grad} \mathcal{L}(\boldsymbol{x}_t)$ must converge to $0$. Summing over $t$ of Equation (11), we have

$$
\sum_{t \geq 0} \mathbb{E}_z[\mathcal{L}(\boldsymbol{x}_{t+1}) \mathcal{L}(\boldsymbol{x}_t) \mid \mathcal{F}_t] \leq \sum_{t \geq 0} 256\eta_t^2 - \sum_{t \geq 0} \xi \|\operatorname{grad} \mathcal{L}(\boldsymbol{x}_t)\|^2.
$$

From Equation (11), we know that $\mathcal{L}(\boldsymbol{x}_t)$ satisfies the assumption in Lemma 2. Hence, $\sum_{t \geq 0} \mathbb{E}_z[\mathcal{L}(\boldsymbol{x}_{t+1}) - \mathcal{L}(\boldsymbol{x}_t) \mid \mathcal{F}_t]$ converges almost surely, implying $\sum_{t \geq 0} \xi \|\operatorname{grad} \mathcal{L}(\boldsymbol{x}_t)\|^2$ also converges almost surely. Combining with the fact that $\operatorname{grad} \mathcal{L}(\boldsymbol{x}_t)$ converges almost surely, which we have proved, we show that $\operatorname{grad} \mathcal{L}(\boldsymbol{x}_t)$ must converge to $0$. $\qquad\square$

**Theorem 2.** When the update rule given by Equation (7) is applied to $\mathcal{L}(\boldsymbol{x})$, and the learning rate satisfies the usual condition in stochastic approximation, *i.e.*, $\sum_t \eta_t^2 < \infty$ and $\sum_t \eta_t = \infty$, $\boldsymbol{x}$ converges almost surely to a critical point $\boldsymbol{x}^*$ and $\operatorname{grad} \mathcal{L}(\boldsymbol{x})$ converges almost surely to $0$, *i.e.*,

$$
\Pr\left(\lim_{t \to \infty} \mathcal{L}(\boldsymbol{x}_t) = \mathcal{L}(\boldsymbol{x}^*)\right) = 1, \quad \Pr\left(\lim_{t \to \infty} \operatorname{grad} \mathcal{L}(\boldsymbol{x}_t) = 0\right) = 1.
$$

*Proof.* Let
$$\boldsymbol{x}_{t+1}^{\exp} = \exp_{\boldsymbol{x}_t}\left(-\eta_t\left(1 + \frac{\boldsymbol{x}_t^\top \nabla f(\boldsymbol{x}_t)}{\|\nabla f(\boldsymbol{x}_t)\|}\right)\left(I - \boldsymbol{x}_t\boldsymbol{x}_t^\top\right)\nabla f(\boldsymbol{x}_t)\right)$$
be the updated point mapped via exponential mapping.

Let
$$\boldsymbol{x}_{t+1} = R_{\boldsymbol{x}_t}\left(-\eta_t\left(1 + \frac{\boldsymbol{x}_t^\top \nabla f(\boldsymbol{x}_t)}{\|\nabla f(\boldsymbol{x}_t)\|}\right)\left(I - \boldsymbol{x}_t\boldsymbol{x}_t^\top\right)\nabla f(\boldsymbol{x}_t)\right)$$
be the updated point mapped via retraction.

The retraction is a first-order approximation of the exponential mapping, *i.e.*, $\exists M > 0$ such that $d(R_{\boldsymbol{x}_t}(\epsilon\boldsymbol{\alpha}), \exp_{\boldsymbol{x}_t}(\epsilon\boldsymbol{\alpha})) < M\epsilon^2$ for $\epsilon > 0$ sufficiently small, where $\|\boldsymbol{\alpha}\| = 1$.

Then,
$$\mathcal{L}(\boldsymbol{x}_{t+1}) - \mathcal{L}(\boldsymbol{x}_t) \le |\mathcal{L}(\boldsymbol{x}_{t+1}) - \mathcal{L}(\boldsymbol{x}_{t+1}^{\exp})| + \mathcal{L}(\boldsymbol{x}_{t+1}^{\exp}) - \mathcal{L}(\boldsymbol{x}_t), \tag{12}$$
where $\mathcal{L}(\boldsymbol{x}_{t+1}^{\exp}) - \mathcal{L}(\boldsymbol{x}_t)$ is proved to be bounded in Equation (10) of Lemma 3, and the term $|\mathcal{L}(\boldsymbol{x}_{t+1}) - \mathcal{L}(\boldsymbol{x}_{t+1}^{\exp})|$ is also bounded

$$|\mathcal{L}(\boldsymbol{x}_{t+1}) - \mathcal{L}(\boldsymbol{x}_{t+1}^{\exp})| \le 256M\eta_t^2.$$

by applying a similar derivation as in Lemma 3 from Equation (9) to Equation (10).

Therefore, $\mathcal{L}(\boldsymbol{x}_t)$ is a quasi-martingale and converges almost surely according to Lemma 2. Also, $\sum_{t=1}^{\infty} \xi \|\text{grad}\,\mathcal{L}(\boldsymbol{x}_t)\|^2 < \infty$ almost surely, which means $\|\text{grad}\,\mathcal{L}(\boldsymbol{x}_t)\|$ can only converge to 0 if it converges because $\sum_t \eta_t = \infty$.

Finally, to prove $\text{grad}\,\mathcal{L}(\boldsymbol{x}_t)$ almost surely converges, we repeat the above proof by replacing $\mathcal{L}(\boldsymbol{x}_t)$ with $\|\text{grad}\,\mathcal{L}(\boldsymbol{x}_t)\|^2$ so that we can arrive at a similar form of Equation (12) and use the same procedure to show $\|\text{grad}\,\mathcal{L}(\boldsymbol{x}_t)\|^2$ is a quasi-martingale and converges almost surely. $\square$