[Reviews · NeurIPS 2019]

Reviewer 1



The paper proposes a new approach for unsupervised learning of text embeddings. The method is grounded on theoretical results and the proposed method seems efficient (faster training) according to the experimental results. The big problem with the paper is the experimental setup/results: 1) Experiments are performed with embeddings of size 100 only. Methods such as word2vec are known to perform better in analogy experiments when using embedding dimension of 300 or more. 2) The authors mention that they achieve state-of-the-art results for word similarity and analogy tasks. However, the authors do not compare their numbers with the ones reported in previous works, they only compare with their own runs of previous embedding models. For instance, the numbers reported in the original fastText paper (https://aclweb.org/anthology/Q17-1010) are much better than the ones reported in the experimental section. In the word analogy task, (the original) FastText achieves accuracy of 77.8 (SemGoogle) and 74.9 (SynGoogle), which are significatively better than the numbers reported for JoSE. Is this only related to the embedding size? The original FastText uses embedding size of 300. Does JoSE improve when the embedding size increases? Is it still significantly superior to the baselines when using embedding size of 300? Are embeddings of larger dimensionality less sensitive to the effect of training on Euclidean space and testing on spherical space? 3) Although the clustering task is interesting, I believe that a text ranking task such as answer selection (that is normally tackled using cosine similarity on embeddings) would give good insights about the effectiveness of the paragraph embeddings for a more real-world task. === Thank you for adding the new results. I've change my score to 6.

Reviewer 2



This paper proposes JoSE, a method to train word embeddings. Their unsupervised approach is rooted in the principle that words with similar contexts should be similar, where they have some novelty in their generative model using both word-word and word-paragraph embeddings and the novelty largely lies in their constraint that all embeddings are on the unit sphere - where they derive an optimization procedure for this constrained problem using Riemannian optimization. They also utilize word, paragraph s The empirical results form this paper are strong - outperforming the GloVe, Poincare Glove, and Word2vec baselines considerably in some cases. FastText is also outperformed as well, though less so, but FastText does have the advantage of using character n-gram information which is not used in JoSE. They also evaluate on analogies and embedding documents from the 20 newsgroups dataset and clustering them, evaluating on the purity of the clusters. While using spherical topology for embeddings is not anything new, I have not seen it applied in this manner. I find the results impressive enough for acceptance - and I checked the paper for experimental issues that would give them an advantage. They do have a hyperparameter they tune but they use the same value for all experiments. Other methods keep default hyperparameters. One concern I also had was efficiency, but their method is actually the fastest of the methods they compare to as well. A couple minor nits. SIF is used as a baseline, what does that mean exactly? SIF largely corresponds to techniques to leverage existing embeddings using SVD etc. Was that done here? Or by SIF do you mean the SIF embecdings were downloaded? If the latter, "Towards Universal Paraphrastic Sentence Embeddings" should be cited since those are largely the actual embeddings in SIF.

Reviewer 3



- This paper proposes a novel text embedding approach in the spherical space and develops an optimization algorithm to train the model. - The claims in this paper are well supported by theories and their derivations. - The results on text embedding applications look pretty good. It shows the model can learn text embeddings in the spherical space effectively and efficiently. Weaknesses: - The paper is well organized, however it might be hard for the readers to reproduce the results without the code, especially in Section 4. A step to step illustration of the optimization algorithm would help. - The authors did not show the possible weaknesses of the proposed model.

[Author Response · NeurIPS 2019]

We sincerely thank the reviewers for their thoughtful comments. To address the major comments, for lack of space, we
only present the performances of **JoSE**-joint (denoted as **JoSE**) because it generally performs better than **JoSE**-base.
(**Reviewer 1**) **Q1**: Show results of 300-d embeddings and compare them with those reported in previous work. **A1**: We
show the performance of our model and baseline models with 200-d and 300-d embeddings (Table 1). We obtained
similar but not same results as in fastText paper, probably because the wikipedia dump changes over time.

**Q2**: Test JoSE with different embedding sizes and explain the results. **A2**: We used 100-d embeddings as evaluation in
the paper because they are efficient to learn and usually sufficient in our tasks, especially the word similarity task. That
said, our model also benefits from higher embedding dimensions. We observe the following: (1) **JoSE** almost constantly
outperforms all the baselines except fastText, which incorporates subword information. Our framework can also be
improved by leveraging subword information (as future work). (2) 100-d **JoSE** achieves comparable performances
with 300-d Word2Vec/GloVe, but its performance increases marginally when $d$ goes higher. A recent work[1] shows that
different Euclidean word embedding algorithms have different sensitivities to dimensionality, and higher dimension
does not necessarily lead to better performance. We will do further study on dimensionality sensitivity and optimal
dimensionality selection for non-Euclidean embedding.

Table 1: Spearman rank correlation on word similarity & Accuracy on word analogy.

| Dimension | Model | Similarity | | | | | Analogy | | |
|---|---|---|---|---|---|---|---|---|---|
| | | WordSim353 | MEN | Simlex | MTurk | RW | SemGoogle | SynGoogle | MSR |
| 200 | Word2Vec | 0.652 | 0.687 | 0.329 | 0.671 | 0.437 | 0.717 | 0.628 | 0.526 |
| | GloVe | 0.611 | 0.655 | 0.316 | 0.665 | 0.441 | 0.705 | 0.591 | 0.498 |
| | fastText | 0.703 | 0.717 | 0.334 | 0.685 | **0.464** | **0.728** | 0.674 | 0.540 |
| | Poincaré GloVe | 0.641 | 0.671 | 0.324 | 0.667 | 0.444 | 0.698 | 0.612 | 0.512 |
| | **JoSE** | **0.730** | **0.728** | **0.347** | **0.690** | 0.459 | 0.725 | **0.675** | **0.555** |
| 300 | Word2Vec | 0.719 | 0.717 | 0.336 | 0.678 | 0.455 | 0.780 | 0.709 | 0.563 |
| | GloVe | 0.648 | 0.704 | 0.331 | 0.660 | 0.438 | 0.716 | 0.609 | 0.500 |
| | fastText | 0.710 | 0.727 | 0.338 | 0.682 | **0.498** | **0.782** | **0.746** | **0.630** |
| | Poincaré GloVe | 0.667 | 0.715 | 0.335 | 0.669 | 0.455 | 0.707 | 0.627 | 0.516 |
| | **JoSE** | **0.733** | **0.735** | **0.358** | **0.694** | 0.465 | 0.775 | 0.716 | 0.583 |

(**Reviewer 2**) **Q3**: Conduct qualitative analysis and explain why training helps. **A3**: We present the vector dot
product and cosine similarity between the two words in pair A: *journey-voyage* and B: *baby-mother* in Table 2
using Word2Vec and **JoSE**. In WordSim353, pair A has higher ground truth similarity than pair B. During training,
Word2Vec assigns higher dot product to pair A by increasing the vector norms of words. However, the cosine
similarity of pair A is still smaller than pair B. The gap between training space and usage space leads to wrong
relative ranking of the two pairs. **JoSE** closes this gap and ranks two pairs consistently during training and testing.

**Q4**: Show performance on downstream tasks. **A4**: We will include
a text ranking task as suggested by Reviewer 1. Word embedding
also has its niches despite the effectiveness of contextualized word
representations from deep language models. Many text mining tasks
require context-free (static) word representations. For example,
query expansion[2] and text concept set retrieval[3] expand initial user
query or seed term set (usually consists of a few words) by retrieving
similar words in the embedding space for semantic enrichment. In the aforementioned tasks, word similarity is directly
employed. Since our models achieve state-of-the-art performance on word similarity evaluation, the benefit will carry
over to the downstream tasks. Moreover, large-scale ad-hoc searching and recommendation systems require high
efficiency, where our model has great advantage over deep language models.

Table 2: Dot product & cosine sim. of word pairs.

| Model | A: *journey-voyage* | | B: *baby-mother* | |
|---|---|---|---|---|
| | Dot | Cos | Dot | Cos |
| Word2Vec | 6.710 | 0.694 | 4.813 | 0.717 |
| **JoSE** | 0.750 | 0.750 | 0.647 | 0.647 |

**Q5**: Meaning of SIF. **A5**: SIF refers to a baseline model (citation [2] in our original submission). We will also include
the citation of "Towards Universal Paraphrastic Sentence Embeddings" in the revision.

(**Reviewer 3**) **Q6**: Show an algorithm table for better understanding and reproducibility. **A6**: Thanks. We will include
an algorithm table and make the process flow clearer in the revision. Please also note that we released our code and its
link has been mentioned in the abstract of the submission.

**Q7**: What are word-word and word-paragraph co-occurrence statistics and how they are exploited? **A7**: Word-word
co-occurrence refers to the appearance of word $u$ in the local context window of word $v$; word-paragraph co-occurrence
refers to the appearance of word $u$ in paragraph $d$. In our framework, both statistics are jointly captured by Eq. (3),
where the objective maximizes both word-word co-occurrence probability $p(v \mid u)$ and word-paragraph co-occurrence
probability $p(u \mid d)$ under the spherical generative model. We will make this clearer in the revision.

[1] Z. Yin and Y. Shen. On the dimensionality of word embedding. In NeurIPS, 2018.
[2] F. Diaz, B. Mitra, and N. Craswell. Query expansion with locally-trained word embeddings. In ACL, 2016.
[3] M. Zaheer, S. Kottur, S. Ravanbakhsh, B. Póczos, R. Salakhutdinov, and A. J. Smola. Deep sets. In NIPS, 2017.


[Meta-Review · NeurIPS 2019]

This paper was reviewed by three expert reviewers and received two Weak Accept and one Accept recommendations. After rebuttal, all the three reviewers are positive about this paper, and agree that the proposed model is novel. Initially, R1 had great concerns about the experimental setup/results. The new results provided in the rebuttal successfully solved R1's main concern. On the one hand, both R2 and R3 think the empirical results are strong and impressive. On the other hand, R1 thinks that the improvements are marginal when the embedding size is large. On balance, the AC recommends accepting the paper, but also strongly advise the authors to include additional comparisons and other revisions suggested by the reviewers and promised in the rebuttal.